# Comparative analysis of the characteristics, care pathways, and outcomes of English and Welsh major trauma patients injured by high versus low energy transfer mechanisms in 2019

Thomas A G Shanahan[1,2☯], Michael Tonkins[3☯], Omar Bouamra[4],
Dhushy Surendra Kumar[5], Antoinette Edwards[5], Laura White[5], Anthony Kehoe[6],
Jason E Smith[7], Timothy J Coats[8], Fiona Lecky[9]*

1 Emergency Department, Royal Perth Hospital, Perth, Western Australia, Australia, 2 Faculty of Medicine, Nursing and Health Sciences, Monash University, Clayton, Victoria, Australia, 3 Emergency Department, Sheffield Teaching Hospitals, Sheffield, South Yorkshire, United Kingdom, 4 Trauma Audit and Research Network (TARN), University of Manchester, Manchester, Bury, United Kingdom, 5 National Major Trauma Registry, NHS Arden and Greater Midlands Commissioning Support Unit, Derby, Derbyshire, United Kingdom, 6 Emergency Department, Musgrove Park Hospital, Taunton, Somerset, United Kingdom, 7 Emergency Department, University Hospitals Plymouth NHS Trust, Plymouth, Devon, United Kingdom, 8 Department of Cardiovascular Sciences, University of Leicester, Leicester, Leicestershire, United Kingdom, 9 Centre for Urgent and Emergency Care Research (CURE), Health Services Research, School of Medicine and Population Health, University of Sheffield, Sheffield, South Yorkshire, United Kingdom

☯ These authors contributed equally to the work and can both put themselves as first author on this work.
* f.e.lecky@sheffield.ac.uk

## Abstract

### Background

Recent trends in high-income countries indicate a shift in the causes of major trauma, with low-energy transfer mechanisms, particularly falls from less than two meters, becoming increasingly prevalent. This study aimed to compare the demographics, care processes, and outcomes of major trauma patients injured by low and high-energy transfer mechanisms.

### Methods

This comparative cohort study utilized anonymized data from adult patients recorded in the Trauma Audit and Research Network in 2019. Patients were categorized into low-energy (falls less than 2 meters) and high-energy (other mechanisms) groups. The study focused on patients with an Injury Severity Score (ISS) greater than 15. Data from up to 179 English and Welsh hospitals were included.

### Results

In 2019, 53.6% (n = 16,087) of major trauma patients were injured by low-energy falls. When compared to the high-energy cohort, these affected older patients (median

**Data availability statement:** Data cannot be shared as both: (i) the Data Sharing Agreements established between the TARN (now NMTR) and member NHS Trusts, and (ii) the Section 251 HRA approval for analysis of anonymized TARN data specify the need for data access agreements with third parties. Proposals to access the study data, data dictionary, analytic code, and analysis scripts may be submitted to: queries@hra.nhs.uk. Proposals are subject to review by the TARN Research committee. A Data Transfer Agreement is required, and all access must comply with TARN HRA approval.

**Funding:** TARN (now NMTR) part remunerates FL, TC and DK in their roles as Research Director, Chair and Audit Director, respectively. AE and LW are full time employees of TARN (now NMTR) (Executive and Operations Directors). Participating NHS Trusts funds TARN. TAGS and MT, academic clinical fellowships are funded by Health Education England (HEE) / NIHR for this research project. The views expressed in this publication are those of the author(s) and not necessarily those of the NIHR, NHS or the UK Department of Health and Social Care. no other relationships or activities that could appear to have influenced the submitted work. All authors are independent from funders and that all authors, external and internal, had full access to all the data (including statistical reports and tables) in the study and can take responsibility for the integrity of the data and the accuracy of the data analysis is also required.

**Competing interests:** All authors have completed the ICMJE uniform disclosure form atwww.icmje.org/coi_disclosure.pdf and declare: TARN (now NMTR) part remunerates FL, TC and DK in their roles as Research Director, Chair and Audit Director, respectively. AE and LW are full time employees of TARN (Executive and Operations Directors). Participating NHS Trusts funds TARN (now NMTR). TAGS and MT were academic clinical fellowship when the work was conducted and that was funded by Health Education England (HEE) / NIHR for this research project. The views expressed in this publication are those of the author(s) and not necessarily those of the NIHR, NHS or the UK

age 80 vs. 47 years; $p < 0.001$), with a higher prevalence of pre-existing comorbidities (90.4% [95%CI 89.9–90.8] vs. 56.2% [95%CI 55.4–57.0]; $p < 0.001$) and traumatic brain injuries (74.0% [95%CI 73.3–74.7] vs. 49.8% [95%CI 48.9–50.6]; $p < 0.001$). Low-energy fall patients were more likely to be initially treated in Trauma Units rather than Major Trauma Centres and received fewer interventions such as surgery and critical care admission. Low-energy falls patients had a higher in-hospital mortality rate (14.0% [95%CI 13.5% − 14.6%] vs. 10.3% [95%CI 9.8% − 10.8%]; $p < 0.0001$).

## Conclusions

The increasing burden of major trauma from low-energy falls necessitates a re-evaluation of current trauma care systems and injury prevention strategies to better serve this distinct and growing patient population. Future research should focus on optimizing care pathways, defining patient orientated outcomes and improving outcomes for patients injured by low-energy falls.

## Introduction

Classifying injuries by energy transfer has received growing attention recently by trauma systems [1–3]. Globally, injuries sustained through high-energy transfer mechanisms, such as road traffic collisions are ranked as the top cause of disability-adjusted life-years lost for people aged 10–49 years [4]. However, recent studies in high income countries have identified injuries from low-energy transfer mechanisms, such as falls less than two metres, as an increasingly common cause of major trauma presenting to hospital [2,5–7]. Between 2012–2017, falls from less than two meters accounted for 57.8% (n = 177,623) of all major trauma cases with an Injury Severity Score (ISS) >8 in the Trauma Audit and Research Network (TARN) in England and Wales [8]. These low-energy falls occur predominately in older people (over 64 years) [9,10] who are more likely to fall due to factors such as ageing, frailty, multiple co-morbidities, polypharmacy, and cognitive decline [11].

In 2018 we reported profound changes in the demographics of recorded major trauma cases in the largest European trauma registry between 1990 and 2013, with an increase in the mean age of major trauma patients (from 36 to 53 years) and a decline in male preponderance (73.0% to 69.0%) [12]. Low-energy falls replaced road traffic collisions as the commonest mechanism of injury [9].

Accurate and up to date understanding of the characteristics and volume of major trauma patients is necessary for effective injury prevention and for developing optimal trauma care systems – both of which are currently designed to respond to those injured by high-energy transfer mechanisms. Therefore, the aim of our study was to compare the characteristics, care processes and outcomes of patients with severe injuries caused by low and high-energy transfer mechanisms to determine whether the current model of care is appropriate.

Department of Health and Social Care. no other relationships or activities that could appear to have influenced the submitted work.

## Methods

### Study design and setting

We conducted a comparative cohort study of patients within the TARN registry in 2019 categorised as either high-energy or low-energy by injury mechanism [2]. Low-energy was defined as falls from less than two metres. High-energy was all other mechanisms, such as road traffic collisions and falls from more than 2 meters. The study design and reporting followed the STROBE guidance for observational studies [13]. The study size was determined by the number of eligible patients included in the TARN database.

### Participants

The inclusion criteria for the TARN registry are as follows: patients of all ages who sustain injury resulting in hospital admission over three days, critical care admission, transfer to a tertiary/specialist centre or death within 30 days of hospital arrival. Isolated femoral neck or single pubic ramus fracture in patients [over] 65 years and simple isolated injuries are excluded. After study inclusion, a dataset of prospectively recorded variables covering demographics plus injury-related physiological, investigation, treatment and outcome parameters is collated using a standard web-based case record form by trained TARN hospital audit co-ordinators. Injury descriptions from imaging, operative and necropsy reports are submitted by TARN co-ordinators. All injuries are coded centrally using the Abbreviated Injury Scale (AIS) (2008 revision of the 2005 AIS dictionary), which enables calculation of the Injury Severity Score (ISS) [14].

Anonymised data from adult patients (18 years and over) with severe injuries (ISS > 15) admitted between January 1, 2019, and December 31, 2019, were collated for analysis. The study was closed at the end of 2019 to avoid the confounding impact of the COVID-19 lockdown on the major trauma disease burden and patient outcomes, which has been studied elsewhere [15].

### Data sources and variables

The following variables were extracted from the TARN database for analysis:

1. Characteristics: gender; age; pre-existing medical conditions.

2. Vital signs on arrival into ED: heart rate, systolic blood pressure; Glasgow Comma Scale (GCS); oxygen saturation.

3. Injuries: Penetrating or blunt; ISS; injury regions with AIS 3 +.

4. Pathway: defined as the level of trauma hospital involvement: first MTC hospital; MTC any time to include transfers from other hospitals into an MTC.

5. Interventions: prehospital doctor; seen by a consultant in ED; Tranexamic acid (TXA) given; TXA given for those given blood; intubation; time to CT; had operation within 24 hours; time to surgery; critical care admission

6. Patient outcomes: Acute care length of stay (LOS); intensive care unit (ICU) LOS; mortality at 30 days post admission or discharge whichever was earlier.

Variables are all extracted from the clinical record (demographics, vital signs and outcomes) only AIS and ISS are derived, and pre-existing medical conditions used the modified Charleson Comorbidity Index [16].

## Statistical methods

We carried out a comparative analysis by energy transfer level. Continuous and ordinal variables (age, time intervals, GCS, AIS, ISS) are presented as median and interquartile range (IQR). Categorical variables are presented as number and percentage. Chi-squared tests were used to compare categorical variables between low-energy and high-energy categories. Bonnet-Price test was used for differences of median for continuous variables.

In the analysis of hospital mortality, mortality at 30 days or at discharge (whichever occurred first) was used as the dependent variable, with admission year as an independent exposure. For the missing values of the arrival GCS, a previously reported imputation technique based on chained equations and Rubin's rules was used with the assumption that the mechanism of missingness is at random (MAR) that is the missing value depends on the measured variables. The imputation contained all the variables included in the study and the outcome [12].

## Patient and public involvement

The TARN research committee approves all study proposals using TARN data and reports twice yearly to the TARN Board. TARN Board membership includes patient and public representation. There was no specific patient and public involvement in this analysis.

## Ethics

The UK Health Research Authority Patient Information Advisory Group (PIAG) has given approval (Section 20) for TARN analysis of anonymised patient data.

## Results

The study included 30,582 patients. All data was 100% complete other than the Charleson co-morbidity index, which was missing in 2% of patients and GCS, which was missing in 8.2% of patients prior to imputation.

Patients injured by low-energy falls (53.6%, n = 16,087) were older (median age 80 vs. 47 years; p < 0.001), less likely to be male (53.8% vs. 73.8%; p < 0.001), more likely to have pre-injury co-morbidity (90.4% [95%CI 89.8–90.8]vs. 56.2% [95%CI 55.4–57.0]; p < 0.001), less likely to have polytrauma (10.8% [95% 10.3–11.3] vs. 38.2% 95%CI 37.4–39.0]; p < 0.001) and more likely to have a traumatic brain injury visible on CT scan (74.0% [95%CI 73.3–74.7 vs. 49.8% [95%CI 48.9–50.6]; p < 0.001) than patients injured by high-energy transfer (46.4%, n = 14,495) (Table 1). Although most patients in both groups presented with normal or only mildly impaired consciousness (GCS 13–15) the low-energy falls cohort were less than half as likely to present with severe impairment of consciousness (GCS 3–8, 6.0% [95%CI 5.7–6.4] vs. 15.3% [95%CI 14.7–15.9]; p < 0.001) (Table 1). Overall, the anatomical injury severity was lower in the low-energy cohort (median ISS 20 vs. 25; p < 0.001; Table 1).

Patients in the low-energy falls cohort were less likely to initially present to a Major Trauma Centre (MTC) (41.7% [95%CI 40.9–42.4] vs. 67% [95%CI 66.2–67.8]; p < 0.001) and less likely to receive MTC care at any time (49.8% [95%CI 49.1–50.6]vs. 76.6% [95%CI 75.9–77.3]; p < 0.001) than the high-energy transfer cohort (Table 2). They were seen less frequently by a doctor in the prehospital environment (1.4% [95%CI 1.25–1.62] vs. 13.6% [95%CI 13.0–14.1]; p < 0.001) or consultant in the emergency department (37.6% [95%CI 36.9–38.4] vs. 69.1% [95%CI 68.4–69.9]; p < 0.001) (Table 1). The low-energy transfer cohort received tranexamic acid (TXA) less frequently both on admission (5.1% [95%CI 4.8–5.4] vs. 28.4% [95%CI 27.7–29.2]; p < 0.001) and if administered blood products (0.4% [95%CI 0.28–0.48] vs. 7.8% [95%CI

**Table 1. Comparison of major trauma patients injured by low versus high energy transfer in 2019 – all TARN Hospitals: Characteristics; vital signs; and injuries.**

| 2019 | Low energy | High energy | p-values |
|---|---|---|---|
| Total number of patients | 16087 | 14495 | |
| Mean number of patients per hospital | 90 | 82 | |
| Number of hospitals | 179 | 176 | |
| Receiving CT scan, n (%) [95% CI] | 14574 (90.6%) [90.0-91.0] | 12600 (86.9%) [86.4-87.5] | <0.001 |
| **Characteristics** | | | |
| Female, n (%) [95% CI] | 7431 (46.2%) [45.4-47.0] | 3792 (26.2%) [25.4-26.9] | <0.001 |
| Age, median (IQR) | 80 (68–87) | 47 (28–64) | <0.001 |
| 75 yrs and above, n (%) [95% CI] | 10088 (62.7%) [62.0-63.4] | 1882 (13%) [12.4-13.5] | <0.001 |
| Pre-existing medical conditions*, n (%) [95% CI] | 14536 (90.4%) [89.9-90.8] | 8149 (56.2%) [55.4-57.0] | <0.001 |
| **Vital signs** | | | |
| % O2 SATs on arrival, median (IQR) | 97 (95 - 98) | 98 (96 - 100) | 0.999 |
| Heart rate on arrival bpm, median (IQR) | 80 (70 - 93) | 84 (71 - 100) | <0.001 |
| SBP on arrival, median (IQR) | 143 (124 - 162) | 130 (114 - 147) | <0.001 |
| GCS on arrival, median (IQR) | 15 (14 – 15) | 15 (13 – 15) | 0.999 |
| Mild head injury (GCS 13–15), n (%) [95% CI]** | 13327 (82.8%) [82.0-83.0] | 10882 (75.1%) [74.4] | <0.001 |
| Moderate head injury (GCS 9–12), n (%) [95% CI] | 1042 (6.5%) [6.1-6.9] | 1007 (6.9%) [6.5-7.4] | <0.001 |
| Severe head injury (GCS 3–8), n (%) [95% CI] | 973 (6%) [5.7-6.4] | 2222 (15.3%) [14.7-15.9] | <0.001 |
| GCS Not recorded, n (%) | 6 (0.0%) | 29 (0.2%) | |
| **Injuries** | | | |
| Penetrating | 15 (0.1%) [0.05-0.14] | 640 (4.4%) [4.1-4.7] | <0.001 |
| ISS, median (IQR) | 20 (16 – 25) | 25 (18 – 29) | <0.001 |
| Head AIS 3+, n (%) [95% CI] | 11900 (74%) [73.3-74.7] | 7213 (49.8%) [48.9-50.6] | <0.001 |
| Spine AIS 3+, n (%) [95% CI] | 1014 (6.3%) [5.9-6.7] | 1670 (11.5%) [11.0-12.0] | <0.001 |
| Face AIS 3+, n (%) [95% CI] | 21 (0.1%) [0.07-0.19] | 145 (1%) [0.84-1.16] | <0.001 |
| Thorax AIS 3+, n (%) [95% CI] | 3146 (19.6%) [18.9-20.2] | 7252 (50%) [49.2-50.8] | <0.001 |
| Abdomen AIS 3+, n (%) [95% CI] | 298 (1.9%) [1.6-2.2] | 1733 (12%) [11.4-12.5] | <0.001 |
| Pelvis AIS 3+, n (%) [95% CI] | 1051 (6.5%) [6.2-6.9] | 1799 (12.4%) [11.9-12.9] | <0.001 |
| Upper limb AIS 3+, n (%) [95% CI] | 22 (0.1%) [0.08-0.19] | 263 (1.8%) [1.6-2.0] | <0.001 |
| Lower limb AIS 3+, n (%) [95% CI] | 566 (3.5%) [3.2-3.8] | 1553 (10.7%) [10.2-11.2] | <0.001 |
| Polytrauma, n (%) [95% CI] | 1735 (10.8%) [10.3-11.3] | 5542 (38.2%) [37.4-39.0] | <0.001 |

(TARN- Trauma Audit and Research Network, CT – Computed Tomography Scan, MR – Magnetic Resonance Scan, yrs = years, GCS – Glasgow Coma Scale, SBP = Systolic Blood pressure, mmHg = millimetres of mercury, %o2sat = percentage oxygen saturation, bpm-beats per minute, AIS3+=Abbreviated injury Scale injury of at least 3 out of 6 severity)

\* Pre-injury co-morbidities not recorded as present or absent in 2% of patients.

\*\* GCS was missing in 8.2% of the mild head injury category and imputed as described in the methods section.

7.4–8.2]; p < 0.001) (Table 1). Although rates of CT imaging were greater in the low-energy falls cohort (90.6 [95%CI 90.0–91.0] vs. 86.9% [95%CI 86.4–87.5]; p < 0.001) they waited longer from arrival at hospital for imaging (median time to CT or MR 138 vs. 38 mins; p < 0.001), were less likely to require acute endotracheal intubation (6.3% [95%CI 5.9–6.7] vs 22.3% [95%CI 27.7–29.2]; p < 0.001), be admitted to critical care (14.1% [95%CI 13.6–14.7] vs. 40.2% [95%CI 39.0–41.0]; p < 0.001) or require surgery within 24 hours (9.3% [95%CI 8.7–9.8] vs. 23.1% [95%CI 22.5–23.8]; p < 0.001) (Table 2). Times from hospital arrival to surgery were also longer (8.5 vs. 5.7 hours; p < 0.001) (Table 2).

**Table 2. Comparison of major trauma patients injured by low versus high energy transfer in 2019 – all TARN Hospitals: pathway; interventions and outcomes.**

| 2019 | Low energy | High energy | p-values |
|---|---|---|---|
| **Pathway** | | | |
| First Hospital MTC, n (%) [95% CI] | 6706 (41.7%) [40.9-42.4] | 9712 (67%) [66.2-67.8] | <0.001 |
| MTC any time, n (%) [95% CI] | 8018 (49.8%) [49.1-50.6] | 11105 (76.6%) [75.9-77.3] | <0.001 |
| **Interventions** | | | |
| Pre-hospital Doctor, 1st hospital, n (%) [95% CI] | 231 (1.4%) [1.25-1.62] | 1969 (13.6%) [13.0-14.1] | <0.001 |
| Seen by Consultant in ED, n (%) [95% CI] | 6051 (37.6%) [36.9-38.4] | 10018 (69.1%) [68.4-69.9] | <0.001 |
| TXA given, direct admission, n (%) [95% CI] | 819 (5.1%) [4.8-5.4] | 4121 (28.4%) [27.7-29.2] | <0.001 |
| TXA given for those given blood, direct admission, n (%) [95% CI] | 60 (0.4%) [0.28-0.48] | 1129 (7.8%) [7.4-8.2] | <0.001 |
| Intubation, n (%) [95% CI] | 1007 (6.3%) [5.9-6.7] | 3236 (22.3%) [21.6-23.0] | <0.001 |
| Time to CT (mins), 1st hospital, median (IQR) | 138 (62 - 265) | 38 (23 - 83) | <0.001 |
| Had Operation within 24 hrs, n (%) [95% CI] | 1498 (9.3%) [8.7-9.8] | 3355 (23.1%) [22.5-23.8] | <0.001 |
| Time to surgery (hrs), median (IQR) | 8.5 (2.8 - 16.2) | 5.7 (2.3 - 15.1) | <0.001 |
| Critical care admission, n (%) [95% CI] | 2271 (14.1%) [13.6-14.7] | 5823 (40.2%) [39.0-41.0] | <0.001 |
| **Outcomes** | | | |
| LOS, median (IQR) | 9 (4 – 19) | 8 (4 – 17) | <0.001 |
| LOS ICU, median (IQR) | 3 (1 – 9) | 4 (1 – 10) | <0.001 |
| Mortality, n (%) [95% CI] | 2255 (14.0%) [13.5-14.6] | 1493 (10.3%) [9.8-10.8] | <0.001 |

(CT – Computed Tomography Scan, MTC = major trauma centre, TXA = Tranexamic Acid, LOS = length of stay, ICU = intensive care unit).

The low-energy falls cohort stayed on average one day (median nine vs. eight days; p < 0.001) longer in hospital but had shorter critical care stays (median three vs. four days; p < 0.001). Unadjusted hospital mortality was 3.7% greater in the low-energy falls cohort (14.0% [95%CI 13.5% − 14.6%] vs. 10.3% [95%CI 9.8% − 10.8%]; p < 0.0001) (Table 2).

Due to the large number of patients included in the analysis all the differences between the low and high-energy cohorts were statistically significant apart from GCS on arrival and oxygen saturation.

## Discussion

### Summary of findings

In 2019, low-energy falls accounted for 53.6% of the most severely injured major trauma patients in the national major trauma registry. This cohort is substantially older, suffers more comorbidities, and presents with less physiological derangement. They suffer a greater proportion of head and thoracic injuries, but their injury severity score was lower. They were more likely to be treated in TUs, less likely to receive consultant-led care, had longer waits for imagining, fewer interventions, and higher unadjusted mortality.

### Strengths and limitations of the study

Although many high-income countries have seen similar trends and the pattern of ageing varies between countries, our results may not be internationally generalisable. The large number of patients in each cohort is a strength but it is also means it is difficult to interpret the results as even trivial differences between the cohorts may be statistically significant. TARN case ascertainment at TUs is less complete than at MTCs, and therefore our results may include a degree of bias in the direction of under-reporting low-energy major trauma at TUs.

## Comparison with existing literature

Our analysis confirms the large burden of major trauma from low-energy falls and the predominance of severe head injuries, particularly in patients over 64 years of age, found by this group in England and Wales [9,17]. These findings align with previous work showing that TBI is the most common injury in older trauma patients [9]. The results are consistent with a previous study [2], which found differences in care pathways for TBI patients injured by low and high-energy mechanisms and argued for a different approach to improve prevention and care for TBI patients injured through falls less than two metres [2]. In that study low energy falls did not independently predict ward, ICU or TBI mortality once the confounders of age, injury severity score, GCS, pupillary responses, co-morbidity and anticoagulant status were adjusted for in the multivariable logistic regression model [2]. We did not repeat this analysis in our cohort as TARN did not comprehensively record an important mortality confounder – pre-injury antithrombotic medication status – until the end of 2019. As TBI is the predominant injury in both cohorts the result likely would be similar.

The observation that in 2019 most of the low-energy major trauma cohort (58.3% versus 33.0% of the high-energy cohort) were first treated in a TU and were less likely to receive consultant review suggests that current pre-hospital triage tools (which incorporate high energy mechanism of injury and deranged physiological parameters) are not sufficiently sensitive to detect the severity of their injuries. This is consistent with the findings of the CENTER TBI [2]. The challenge of accurately triaging older patients is well recognised [18–24], and the poor accuracy of all current methods was the finding of a recent systematic review [24]. This may be explained by the different pathophysiology of traumatic brain injury in the two cohorts, highlighted by the greater likelihood of intracranial injury in the low-energy cohort that is not reflected in the initial GCS (Table 1).

The findings of CENTER TBI suggest that acute sub-dural haematoma (ASDH) is the predominant CT finding in the low energy cohort with head injury (49% of the low energy TBI cohort with injuries on CT brain – 18% described as "large") reporting that most high and low energy patients with large ASDH received a craniotomy [2]. However, CENTER TBI was conducted solely in specialist neuroscience centres – and did not include patients from referring hospitals who were not accepted for transfer. Our "whole system" findings suggest that most low energy major trauma patients with positive CT brain scans after head injury are not accepted for neuroscience/ MTC secondary transfer.

Other studies on TARN patients injured in low falls (2015–2019) suggest that initial ambulance transport to the nearest hospital emergency department rather than a more distant specialist major trauma centre does not adversely impact outcomes providing later selective secondary transfer is available [25]. Furthermore, in older adults with ISS > 15 – most of whom are injured in low-energy falls – a frailty score and comprehensive geriatric assessment are more important components of care than the immediate availability of surgical specialists [26,27].

## Interpretation of the study and implications

The growing burden of major trauma resulting from low-energy falls is not new, nor is the fact that these patients have difference characteristics compared to patients injured by high-energy transfer mechanism. However, this study updates our understanding that patients with major trauma caused by low-energy falls are older and present with more normal physiology, despite having traumatic brain injury and thoracic injuries [9]. These differences seem important in planning the next phase of development of integrated trauma systems.

The lower rates of MTC admission at any time, surgical intervention, and admission to critical care in the low-energy cohort are difficult to interpret. There may be systematic access barriers to higher level care in older patients. However, it seems that these patients have a much lower requirement for the specialist surgical intervention that is provided in MTCs, as most have head injuries that do not receive surgical intervention [9]. This might be due to smaller intracranial haematomas requiring evacuation [2]. Moreover, it has already been demonstrated that the ISS had only a low-to-moderate correlation with the need for life-saving interventions [28]. Our previous study [25] showing similar risk adjusted hospital

mortality in low fallers in major trauma centres versus trauma units is limited to a short-term outcome. Longer follow up for patient reported outcomes in future studies is needed to determine whether the current less intense care pathway for older fallers with severe injuries is appropriate or represents therapeutic nihilism.

The low-energy transfer trauma group may therefore need much more emphasis on multidisciplinary geriatric assessment, care and rehabilitation [26,27]. This means that the most appropriate treatment location in future should be driven by the availability of these services rather than trauma surgery. In some places multi-disciplinary geriatric trauma care expertise will be concentrated in MTCs, but in some systems local hospitals will have all the skills required, so MTC transfer will not be required. This multi-disciplinary care and rehabilitation has already been shown to improve outcomes after hip fracture [29] so the same model could be extended to treat other low-energy transfer patients.

The optimal pathways through the trauma care system have not been well defined and it is likely that patient factors such as frailty (TARN commenced recording clinical frailty scores in mid-2019) and ceiling of care decisions (not currently collected) may be important for future care pathway analyses.

### Future research

There is a pressing international need to examine how well the components of trauma networks and injury prevention initiatives serve the increasing volume of major trauma patients injured through low-energy falls. Present systems of both prevention and treatment of major trauma are based on the historical importance of patients injured through road traffic collisions, penetrating trauma and other high-energy injury mechanisms. Research prioritisation processes in trauma care have highlighted the importance of research to define optimal pathways for modern trauma care given the change in the population [30–32]. Focussing research on early identification of low-energy falls patients that benefit from specialist trauma and neurosurgical input, optimal TBI and chest wall injury management, role of early rehabilitation and better definition of what outcomes are important to patients.

### Conclusion

The aging major trauma population, predominantly injured through low-energy falls and suffering severe head injuries, challenges current trauma system design and resource allocation. Although these patients are less likely to require intensive care and surgical interventions, their increasing numbers necessitate an evolution in trauma care pathways. Future research should define which TBI subtypes in low energy falls benefit from neurosurgical intervention; develop patient-centred outcome measures for low-energy falls patients that integrate functional recovery and quality of life and test care pathways that combine early geriatric assessment with targeted interventions for chest injuries and neurosurgical consultation for high-risk sub-groups.

### Acknowledgments

When this work was first conducted it was under the auspices of TARN, which has subsequently become the National Major Trauma Registry. The lead author (the manuscript's guarantor) affirms that the manuscript is an honest, accurate, and transparent account of the study being reported; that no important aspects of the study have been omitted; and that any discrepancies from the study as originally planned (and, if relevant, registered) have been explained.

### Author contributions

**Conceptualization:** Thomas A G Shanahan, Michael Tonkins, Antoinette Edwards, Laura White, Timothy J Coats, Fiona Lecky.

**Data curation:** Thomas A G Shanahan, Michael Tonkins, Omar Bouamra, Fiona Lecky.

**Formal analysis:** Michael Tonkins, Omar Bouamra.

**Investigation:** Thomas A G Shanahan, Fiona Lecky.

**Methodology:** Omar Bouamra, Fiona Lecky.

**Project administration:** Antoinette Edwards, Laura White.

**Supervision:** Antoinette Edwards, Timothy J Coats, Fiona Lecky.

**Writing – original draft:** Thomas A G Shanahan, Michael Tonkins, Dhushy Surendra Kumar, Antoinette Edwards, Laura White, Anthony Kehoe, Jason E Smith, Timothy J Coats, Fiona Lecky.

**Writing – review & editing:** Thomas A G Shanahan, Michael Tonkins, Omar Bouamra, Dhushy Surendra Kumar, Antoinette Edwards, Laura White, Anthony Kehoe, Jason E Smith, Timothy J Coats, Fiona Lecky.

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
