## [Decision Letter · Decision Letter 0]

3 Jan 2025

Dear Dr. Shanahan,

Thank you for submitting your manuscript to PLOS ONE. After careful consideration, we feel that it has merit but does not fully meet PLOS ONE’s publication criteria as it currently stands. Therefore, we invite you to submit a revised version of the manuscript that addresses the points raised during the review process.

We look forward to receiving your revised manuscript.

Kind regards,

Kathleen Finlayson

Academic Editor

PLOS ONE

4. Your ethics statement should only appear in the Methods section of your manuscript. If your ethics statement is written in any section besides the Methods, please delete it from any other section

Reviewers' comments:

Reviewer's Responses to Questions

**Comments to the Author**

1. Is the manuscript technically sound, and do the data support the conclusions?

Reviewer #1: Partly

Reviewer #2: Yes

2. Has the statistical analysis been performed appropriately and rigorously?

Reviewer #1: Yes

Reviewer #2: Yes

3. Have the authors made all data underlying the findings in their manuscript fully available?

Reviewer #1: No

Reviewer #2: Yes

4. Is the manuscript presented in an intelligible fashion and written in standard English?

Reviewer #1: Yes

Reviewer #2: Yes

Reviewer #1: Thank you for the opportunity to review your paper. Overall, I think it's a straightforward paper that is easy to read and understand. My main comment is that I would like to see some more discussion on what your findings actually mean.

You show that patients injured by low-energy falls are older, have more TBI and higher mortality, but overall receive "lower intensity" care. Do you think this is adequate, or should it change? The death rate in the UK population among 45-49 year olds is something like 2-3 per 1000 population per year, whereas it is 50-70 per 1000 population per year for 80-84 year olds. So maybe the mortality difference is not very surprising?

You state that it is challenging triage older patients, but do you think they should be triaged differently, and if so, why? You acknowledge that some of your findings are difficult to interpret, but I think you can do more to connect your suggestions for future research with your findings. For example, what type of brain injuries do the low-energy falls patients have and can the differences in neurosurgical intervention be justified?

In addition to strengthening the discussion, I have the following comments and suggestions for changes:

- You write "However, recent studies in high income countries have identified injuries from low-energy transfer mechanisms, such as falls less than two metres, as the most common cause of major trauma presenting to hospital.", but your references are only from England and Wales. Do you have any references from other countries?

- Add spaces between sentences and citations, replace for example "Classifying injuries by energy transfer has received growing attention recently by trauma systems(1–3)" with "Classifying injuries by energy transfer has received growing attention recently by trauma systems (1–3)."

- Move the TARN eligibility criteria to the Participants section.

- Please add a Variables section and specify your demographics, care processes and outcomes. From the variables you list under Statistical methods, it looks like you're studying variables that cannot be classified as demographics, care processes or outcomes, for example GCS and ISS.

- Please add a Data sources section and clarify how the variables were measured. For example, "Pre-existing medical conditions", how were they measured?

- Clarify how missing data was handled.

- Clarify how you dealt with different follow-up times.

- Add the total number of patients included in the study to the beginning of the Results section.

- Present the amount of missing data for each variable in the Results section. I suggest you include this information in table 1.

- Please be consistent in how you present confidence intervals and p-values. For example, in one place you write "Patients injured by low-energy falls (53.6%, n=16,087) were older (median age 80 vs. 47 years", but in another place you write "the low-energy falls cohort were less than half as likely to present with severe impairment of consciousness (GCS 3-8, 6.0% vs. 15.3%; p<0.001)". I suggest you include the p-values in all comparisons, or preferably the differences and associated confidence intervals, especially as you write that because of the large numbers more or less all differences are statistically significant.

- Spell out MTC the first time it is used.

- I don't see that you include anything on the differences in the anatomical distribution of injuries between the two groups in the Results section. As far as I can see, it looks like the low-energy falls cohort has more isolated head injuries whereas the high-energy falls cohort has more multiple injuries. I suggest you include something on this in the text.

- The paragraph "The low-energy falls cohort stayed on average one day (median nine vs. eight days) longer in hospital but had shorter critical care stays (median three vs. four days). Hospital mortality was 3.5% greater in the low-energy falls cohort (16.1% [95%CI 15% - 17%] vs. 12.6% [95%CI 12% - 13%]; p=<0.0001) (Table 2)" does not match the table, or did you by mistake use the wrong column headings? Because the columns in the table are reversed compared to the paragraph, and compared to table 1.

Reviewer #2: Really clear message. Most trauma is low velocity. the ageing population mean its growing. Reference to the NHFD work on improving outcomes maybe needed for all frailty fractures may add to the discussion.

**Do you want your identity to be public for this peer review?** For information about this choice, including consent withdrawal, please see our Privacy Policy

Reviewer #1: **Yes:** Martin Gerdin Wärnberg

Reviewer #2: No

---

## [Author Response · Author response to Decision Letter 1]

2 Jul 2025

Response:

Data cannot be shared as both: (i) the Data Sharing Agreements established between the TARN (now NMTR) and member NHS Trusts, and (ii) the Section 251 HRA approval for analysis of anonymized TARN data specify the need for data access agreements with third parties. Proposals to access the study data, data dictionary, analytic code, and analysis scripts may be submitted to: queries@hra.nhs.uk. Proposals are subject to review by the TARN Research committee. A Data Transfer Agreement is required, and all access must comply with TARN HRA approval.

Reviewer comment Authors response

Reviewer 1

(i)You show that patients injured by low-energy falls are older, have more TBI and higher mortality, but overall receive "lower intensity" care. Do you think this is adequate, or should it change?

Thank you for the question. If greater intensity means more specialist surgery, particularly neurosurgery then it is uncertain whether the current level is adequate as low energy major trauma patients cared for initially in a trauma unit do not appear to be disadvantaged.

We comment on this in the discussion under “comparisons to existing literature”:

“Other studies on TARN patients injured in low falls (2015-2019) suggest that initial ambulance transport to the nearest hospital emergency department rather than a more distance specialist major trauma centre does not adversely impact outcomes providing later selective secondary transfer is available (25). Furthermore, in older adults with ISS>15 - most of whom are injured in low-energy falls - a frailty score and comprehensive geriatric assessment are more important components of care than the immediate availability of surgical specialists (26,27).”

And in “interpretation of the study and implications”

“The lower rates of MTC admission at any time, surgical intervention, and admission to critical care in the low-energy cohort are difficult to interpret. There may be systematic access barriers to higher level care in older patients. However, it seems that these patients have a much lower requirement for the specialist surgical intervention that is provided in MTCs, as most have head injuries that do not receive surgical intervention (9). This might be due to smaller intracranial haematomas requiring evacuation (2). Moreover, it has already been demonstrated that the ISS had only a low-to-moderate correlation with the need for life-saving interventions (28).”

“The low-energy transfer trauma group may therefore need much more emphasis on multidisciplinary geriatric assessment, care and rehabilitation (26, 27).”

And in “future research”

“Focussing research on early identification of low-energy falls patients that benefit from specialist trauma and neurosurgical input, optimal TBI and chest wall injury management, role of early rehabilitation and better definition of what outcomes are important to patients.”

(ii) The death rate in the UK population among 45-49 year olds is something like 2-3 per 1000 population per year, whereas it is 50-70 per 1000 population per year for 80-84 year olds. So maybe the mortality difference is not very surprising?

Thank you for the comment. We have edited the manuscript to highlight mortality is unadjusted. While higher baseline mortality in older adults is expected, our study specifically examines outcomes following severe traumatic injuries (ISS>15), not all-cause mortality in the general population.

We agree that the observed difference in mortality for an older population is not surprising - age is an independent predictor of hospital mortality in trauma survival prediction models (12), however the less intense care received by the energy transfer cohort with higher mortality is notable.

(iii) You state that it is challenging triage older patients, but do you think they should be triaged differently, and if so, why?

Thank you this is also uncertain as per our response to your first comment. The systematic review we reference (24) found most tools are not specific or sensitive enough in this area. It is unclear that continuing to focus on prehospital triage of older trauma patients is going to be the best use of resources.

In discussion we further comment:

“This means that the most appropriate treatment location in future should be driven by the availability of these services rather than trauma surgery. In some places multi-disciplinary geriatric trauma care expertise will be concentrated in MTCs, but in some systems local hospitals will have all the skills required, so MTC transfer will not be required. This multi-disciplinary care and rehabilitation has already been shown to improve outcomes after hip fracture (29) so the same model could be extended to treat other low-energy transfer patients.”

(iv) You acknowledge that some of your findings are difficult to interpret, but I think you can do more to connect your suggestions for future research with your findings. For example, what type of brain injuries do the low-energy falls patients have and can the differences in neurosurgical intervention be justified?

Thank you in the discussion we have added:

“The findings of CENTER TBI suggest that acute sub-dural haematoma (ASDH) is the predominant CT finding in the low energy cohort with head injury (49% of the low energy TBI cohort with injuries on CT brain -18% described as “large”) reporting that most high and low energy patients with large ASDH received a craniotomy (2). However, CENTER TBI was conducted solely in specialist neuroscience centres - and did not include patients from referring hospitals who were not accepted for transfer. Our “whole system” findings suggest that most low energy major trauma patients with positive CT brain scans after head injury are not accepted for neuroscience/ MTC secondary transfer”

In the conclusion we have added:

“The aging major trauma population, predominantly injured through low-energy falls and suffering severe head injuries, challenges current trauma system design and resource allocation. Although these patients are less likely to require intensive care and surgical interventions, their increasing numbers necessitate an evolution in trauma care pathways. Future research should define which TBI subtypes in low energy falls benefit from neurosurgical intervention; develop patient-centred outcome measures for low-energy falls patients that integrate functional recovery and quality of life and; test care pathways that combine early geriatric assessment with targeted interventions for chest injuries and neurosurgical consultation for high-risk sub-groups.”

In our earlier responses (i)-(iii) we reflect on the appropriateness of this non interventional approach.

(v) You write "However, recent studies in high income countries have identified injuries from low-energy transfer mechanisms, such as falls less than two metres, as the most common cause of major trauma presenting to hospital.", but your references are only from England and Wales. Do you have any references from other countries?

Thank you for the comment.

We have made the following change:

“However, recent studies in high income countries have identified injuries from low-energy transfer mechanisms, such as falls less than two metres, as an increasingly common cause of major trauma presenting to hospital (2-5).

We have added the following references:

2) Lecky FE, Otesile O, Marincowitz C, Majdan M, Nieboer D, Lingsma HF, et al. The burden of traumatic brain injury from low-energy falls among patients from 18 countries in the CENTER-TBI Registry: A comparative cohort study. PLOS Med. 2021 Sep 14;18(9):e1003761.

3) Cameron PA, et al. Over view of major traumatic injury in Australia – implications for trauma system design. Injury. 2020 Jan; 51(1):114-121

4) Farrow L, et al. Epidemiology of major trauma in older adults within Scotland: A national perspective from the Scottish Trauma Audit Group (STAG). Injury. 2023 Dec;54(12)

5) Spering C, Lefering R, Bouillon B, Lehmann W, von Eckardstein K, Dresing K, et al. It is time for a change in the management of elderly severely injured patients! An analysis of 126,015 patients from the TraumaRegister DGU®. Eur J Trauma Emerg Surg Off Publ Eur Trauma Soc. 2020 Jun;46(3):487–97.

As a result of these additional references, we have re-numbered the references in the manuscript.

(vi) Add spaces between sentences and citations, replace for example "Classifying injuries by energy transfer has received growing attention recently by trauma systems(1–3)" with "Classifying injuries by energy transfer has received growing attention recently by trauma systems (1–3)."

Thanks, we have made those changes throughout the paper as suggested.

(vii) Move the TARN eligibility criteria to the Participants section.

Thanks, we have moved it as suggested. See it on page 7.

(viii) Please add a Variables section and specify your demographics, care processes and outcomes. From the variables you list under Statistical methods, it looks like you're studying variables that cannot be classified as demographics, care processes or outcomes, for example GCS and ISS.

Thanks for the comment. We have adjusted the title to better reflect the variables. We have also added a variables section as per the suggestion. See page 7:

Variables

The following variables were extracted from the TARN database for analysis:

1) Demographics: gender; age; pre-existing medical conditions.

2) Vital signs on arrival into ED: heart rate, systolic blood pressure; GCS; oxygen saturation.

3) Injuries: penetrating; ISS; injury regions with AIS 3+

4) Pathway: defined as the level of trauma hospital involvement: first MTC hospital; MTC any time to include transfers from other hospitals into an MTC.

5) Interventions: prehospital doctor; seen by a consultant in ED; TXA given; TXA given for those given blood; intubation; time to CT; had operation within 24 hours; time to surgery; critical care admission

6) Patient outcomes: Acute care length of stay (LOS); intensive care unit (ICU) LOS; mortality at 30 days post admission or discharge whichever is earlier

(ix) Please add a Data sources section and clarify how the variables were measured. For example, "Pre-existing medical conditions", how were they measured?

Thanks for your comment. They are all extracted from the clinical record (demographics, vital signs and outcomes) only AIS and ISS are derived and PMC using the modified Charlson Co morbidity index (Bouarmra et al EMJ 2015, reference 16), See page 7.

(x) Clarify how missing data was handled.

All data were 100% complete other than the Charlson co-morbidity index which was missing in 2% of patients and the 8.2% missing GCS imputed as per below

Page 8: “For patients with missing GCS an imputation technique based on chained equations and Rubin’s rules was used with the assumption that the mechanism of missingness is at random (MAR) that is the missing value depends on the measured variables. The imputation contained all the variables included in the regression model and the outcome (12).”

(xi) Clarify how you dealt with different follow-up times.

TARN follows patients up to discharge from final acute care hospital. All survivors in acute care hospital at 30 days were called survivors even if subsequently died before acute care discharge (as previously published). As per above we have now clarified this in the variables section

“Acute care length of stay (LOS); intensive care unit (ICU) LOS; mortality at 30 days post admission or discharge whichever is earlier”

(xii) Add the total number of patients included in the study to the beginning of the Results section.

Thanks. We had added the following: “The study included 30,582 patients.” See page 8.

(xiii) Present the amount of missing data for each variable in the Results section. I suggest you include this information in table 1. Thanks for the comment. In the results on page 9 we address this as follows: “All data were 100% complete other than the Charleson co-morbidity index, which was missing in 2% of patients and GCS, which was missing in 8.2% of patients.” Missing GCS was imputed this is described in methods we have added a footnote to table 1 re missing co-morbidity data.

(xiv) Please be consistent in how you present confidence intervals and p-values. For example, in one place you write "Patients injured by low-energy falls (53.6%, n=16,087) were older (median age 80 vs. 47 years", but in another place you write "the low-energy falls cohort were less than half as likely to present with severe impairment of consciousness (GCS 3-8, 6.0% vs. 15.3%; p<0.001)". I suggest you include the p-values in all comparisons, or preferably the differences and associated confidence intervals, especially as you write that because of the large numbers more or less all differences are statistically significant.

Thank you for your comment. We have added in p values in all comparisons.

(xv) Spell out MTC the first time it is used. Thanks, we have done this.

(xvi) I don't see that you include anything on the differences in the anatomical distribution of injuries between the two groups in the Results section. As far as I can see, it looks like the low-energy falls cohort has more isolated head injuries whereas the high-energy falls cohort has more multiple injuries. I suggest you include something on this in the text.

Thanks for the comment. On page 8 the first paragraph of the results highlights what you have mentioned: “Patients injured by low-energy falls (53.6%, n=16,087) were older (median age 80 vs. 47 years), less likely to be male (53.8% vs. 73.8%), more likely to have pre-injury co-morbidity (90.4% vs. 56.2%), less likely to have polytrauma (10.8% vs. 38.2%) and more likely to have a traumatic brain injury visible on CT scan (74.0% vs. 49.8%) than patients injured by high-energy transfer (46.4%, n=14,495) (Table 1).”

(xvii) The paragraph "The low-energy falls cohort stayed on average one day (median nine vs. eight days) longer in hospital but had shorter critical care stays (median three vs. four days). Hospital mortality was 3.5% greater in the low-energy falls cohort (16.1% [95%CI 15% - 17%] vs. 12.6% [95%CI 12% - 13%]; p=<0.0001) (Table 2)" does not match the table, or did you by mistake use the wrong column headings? Because the columns in the table are reversed compared to the paragraph and compared to table 1.

Thank you for spotting this. As suspected, this was a transcription error, and we had the column headings the wrong way round. This has been changed.

Reviewer 2

Really clear message. Most trauma is low velocity. the ageing population mean its growing. Reference to the NHFD work on improving outcomes maybe needed for all frailty fractures may add to the discussion.

Thanks for the overall feedback. In the discussion page 10 paragraph 5 we have the following line which we think addresses your point: “This multi-disciplinary care and rehabilitation has already been shown to improve outcomes after hip fracture (25) so the same model could be extended to treat other low-energy transfer patients.”

---

## [Decision Letter · Decision Letter 1]

5 Aug 2025

Dear Dr. Lecky,

Please consider the reviewers' suggestions, in particular re the need to add further details on your analyses, and/or to justify not using a multivariable analysis to support your findings.

We look forward to receiving your revised manuscript.

Kind regards,

Kathleen Finlayson

Academic Editor

PLOS ONE

Journal Requirements:

Reviewers' comments:

Reviewer's Responses to Questions

**Comments to the Author**

Reviewer #1: (No Response)

Reviewer #3: (No Response)

2. Is the manuscript technically sound, and do the data support the conclusions?

Reviewer #1: Partly

Reviewer #3: Yes

3. Has the statistical analysis been performed appropriately and rigorously?

Reviewer #1: No

Reviewer #3: Yes

4. Have the authors made all data underlying the findings in their manuscript fully available?

Reviewer #1: No

Reviewer #3: No

5. Is the manuscript presented in an intelligible fashion and written in standard English?

Reviewer #1: Yes

Reviewer #3: Yes

Reviewer #1: Thank you for consider my previous feedback. I do have some additional comments:

1. Characteristics actually works quite well in the title, rather than the long "demographics, vital signs, injury

patterns" that you have added. My main point in the previous round was that you study many variables which are not demographics, but these are well captured by "characteristics". I would probably keep the title as "Comparative analysis of the characteristics, care pathways, interventions and outcomes of English and Welsh major

trauma patients injured by high versus low energy transfer mechanisms in 2019."

2. Align the aim with the above comment, it still reads as "demographics, care processes and outcomes" and still doesn't cover the vital signs and injury patterns, for example.

3. Include p-values in the abstract.

4. I've put No in response to the question "Has the statistical analysis been performed appropriately and rigorously?". The reason is that you still do not present differences with appropriate uncertainty (for example 95% CIs), but only p-values, for example (58.3% vs. 33.0%; p<0.001). It would be more informative to see an estimate of the difference with uncertainty.

5. Move the information about the missing data to the beginning of the results section, see STROBE.

Reviewer #3: Overall, I find this an interesting article that draws attention to the changing trauma population. I have a few suggestions for improvement.

- Titel: I prefer a shorter title, e.g. Comparative analysis of the characteristics, care pathways, and outcomes of English and Welsh major trauma patients injured by high versus low energy transfer mechanisms

- Introduction

- The aim is stated as: “Therefore, the aim of our study was to compare the demographics, care processes and outcomes of patients with severe injuries caused by low and high-energy transfer mechanisms.”. I assume that the underlying question is whether the care provided for high-energy trauma is also appropriate or necessary for low-energy trauma cases. I feel this link is missing in the final paragraph of the introduction.

- Method section;

o Please indicate in the methods section which version of the AIS was used.

o Minor detail: In section Participants: “After study inclusion, a dataset of prospectively recorded variables covering demographics plus injury-related physiological, 7 investigation, treatment and outcome parameters are collated” has to be is collated.

o Variables:

1. Pre-existing medical conditions do not fall under the category of demographic data, add a category clinical data or change demographics to patient characteristics.

3. Change penetrating in Penetrating or blunt

- Results: The p-values are described, but not showed in the tables. Please add a column in the tables with p-values.

- Statistical methods:

o Under ordinal variables AIS is mentioned. I assume that it concerns the AIS severity, otherwise it is not ordinal.

o Glasgow Coma Scale can be abbreviated to GCS in the second paragraph.

o I wonder why the authors did not conduct a multivariable analysis to compare the two groups on outcome and adjust for characteristics of patients and injury (age, comorbidity, severity of injury etc)? It is currently difficult to draw firm conclusions about whether low-fall patients are inappropriately receiving higher-level care. It appears that they are less severely injured but older and therefore more likely to be transported to a non-major trauma center (nMTC). It is in line with the comment of reviewer 1 that it is no surprise that the low fall group has a higher mortality. I would strongly advise to do a more sophisticated comparison between high and low falls to add more depth to the manuscript.

**Do you want your identity to be public for this peer review?** For information about this choice, including consent withdrawal, please see our Privacy Policy

Reviewer #1: **Yes:** Martin Gerdin Wärnberg

Reviewer #3: No

---

## [Author Response · Author response to Decision Letter 2]

23 Jan 2026

Reviewer comment Authors response

Reviewer 1

1. Characteristics actually works quite well in the title, rather than the long "demographics, vital signs, injury

patterns" that you have added. My main point in the previous round was that you study many variables which are not demographics, but these are well captured by "characteristics". I would probably keep the title as "Comparative analysis of the characteristics, care pathways, interventions and outcomes of English and Welsh major

trauma patients injured by high versus low energy transfer mechanisms in 2019."

Thanks for your comment. We have changed the title as follows:

“Comparative analysis of the characteristics, care pathways, and outcomes of English and Welsh major trauma patients injured by high versus low energy transfer mechanisms in 2019.”

2. Align the aim with the above comment, it still reads as "demographics, care processes and outcomes" and still doesn't cover the vital signs and injury patterns, for example.

Thanks for your comment. We have removed demographics and replaced it with characteristics as per the title.

“Therefore, the aim of our study was to compare the characteristics, care processes and outcomes of patients with severe injuries caused by low and high-energy transfer mechanisms.”

3. Include p-values in the abstract.

Thanks for your comment. We have included the p-values and 95% CI for in-hospital mortality.

“Results: In 2019, 53.6% (n=16,087) of major trauma patients were injured by low-energy falls. When compared to the high-energy cohort, these affected older patients (median age 80 vs. 47 years; p<0.001), with a higher prevalence of pre-existing comorbidities (90.4% [95%CI 89.9-90.8] vs. 56.2% [95%CI 55.4-57.0]; p<0.001) and traumatic brain injuries (74.0% [95%CI 73.3-74.7] vs. 49.8% [95%CI 48.9-50.6]; p<0.001). Low-energy fall patients were more likely to be initially treated in Trauma Units rather than Major Trauma Centres and received fewer interventions such as surgery and critical care admission. Low-energy falls patients had a higher in-hospital mortality rate (14.0% [95%CI 13.5% - 14.6%] vs. 10.3% [95%CI 9.8% - 10.8%]; p<0.0001).

4. I've put No in response to the question "Has the statistical analysis been performed appropriately and rigorously?". The reason is that you still do not present differences with appropriate uncertainty (for example 95% CIs), but only p-values, for example (58.3% vs. 33.0%; p<0.001). It would be more informative to see an estimate of the difference with uncertainty.

We have added 95%CI for all data in tables 1 and 2 and in the manuscript.

5. Move the information about the missing data to the beginning of the results section, see STROBE.

Thanks for your comment. We have modified as suggested.

“The study included 30,582 patients. All data was 100% complete other than the Charleson co-morbidity index, which was missing in 2% of patients and GCS, which was missing in 8.2% of patients prior to imputation. “

Reviewer 3

Title: I prefer a shorter title, e.g. Comparative analysis of the characteristics, care pathways, and outcomes of English and Welsh major trauma patients injured by high versus low energy transfer mechanisms

Thanks for your comment. We have modified the title as per both review comments as follows:

“Comparative analysis of the characteristics, care pathways, and outcomes of English and Welsh major trauma patients injured by high versus low energy transfer mechanisms in 2019.”

The aim is stated as: “Therefore, the aim of our study was to compare the demographics, care processes and outcomes of patients with severe injuries caused by low and high-energy transfer mechanisms.”. I assume that the underlying question is whether the care provided for high-energy trauma is also appropriate or necessary for low-energy trauma cases. I feel this link is missing in the final paragraph of the introduction.

Thanks for your comment. We have modified as follows:

“Therefore, the aim of our study was to compare the characteristics, care processes and outcomes of patients with severe injuries caused by low and high-energy transfer mechanisms to determine whether the current model of care is appropriate.”

Please indicate in the methods section which version of the AIS was used.

We now indicate that “the 2008 revision of the 2005 AIS dictionary” was used in method

Minor detail: In section Participants: “After study inclusion, a dataset of prospectively recorded variables covering demographics plus injury-related physiological, 7 investigation, treatment and outcome parameters are collated” has to be is collated.

Thanks, we have changed are to is.

Pre-existing medical conditions do not fall under the category of demographic data, add a category clinical data or change demographics to patient characteristics.

Thanks for your comment. We have changed to characteristics throughout.

Change penetrating in Penetrating or blunt

Thanks, we have changed accordingly.

Results: The p-values are described but not showed in the tables. Please add a column in the tables with p-values.

Thanks for your comment. We had put a * to explain the p-values in the submitted manuscript. But have now added a column to the table as suggested with p-values.

Under ordinal variables AIS is mentioned. I assume that it concerns the AIS severity, otherwise it is not ordinal.

Yes, AIS greater than 3 was used and reported in table 1.

Glasgow Coma Scale can be abbreviated to GCS in the second paragraph.

Thanks, we have changed accordingly.

I wonder why the authors did not conduct a multivariable analysis to compare the two groups on outcome and adjust for characteristics of patients and injury (age, comorbidity, severity of injury etc)? It is currently difficult to draw firm conclusions about whether low-fall patients are inappropriately receiving higher-level care. It appears that they are less severely injured but older and therefore more likely to be transported to a non-major trauma centre (nMTC). It is in line with the comment of reviewer 1 that it is no surprise that the low fall group has a higher mortality. I would strongly advise to do a more sophisticated comparison between high and low falls to add more depth to the manuscript.

Thank you, the main focus of the paper is a direct comparison of the characteristics, care pathways and outcomes. We considered conducting such a multivariable analysis but have not done so as previous research has shown that once the relevant patient and injury confounders are adjusted for the higher mortality in the low energy cohort is no longer a significant association. We now allude to this in “discussion – comparison to existing literature”

Our analysis confirms the large burden of major trauma from low-energy falls and the predominance of severe head injuries, particularly in patients over 64 years of age, found by this group in England and Wales (9, 17). These findings align with previous work showing that TBI is the most common injury in older trauma patients (9). The results are consistent with a previous study (2), which found differences in care pathways for TBI patients injured by low and high-energy mechanisms and argued for a different approach to improve prevention and care for TBI patients injured through falls less than two metres (2). In this study low energy falls did not independently predict ward or ICU or TBI mortality once the confounders of age, ISS, GCS, pupillary responses, co-morbidity and anticoagulant status were adjusted for in the multivariable logistic regression model (2). We did not repeat this analysis in our cohort as TARN did not comprehensively record an important mortality confounder - pre-injury antithrombotic medication status - until the end of 2019. As TBI is the predominant injury in both cohorts the result likely would be similar.

We have also added further to the “interpretation of findings “ section of discussion” to address this point made by the reviewer about the appropriateness of care received:

“Our previous study showing similar risk adjusted hospital mortality in low fallers in major trauma centres versus trauma units is limited to a short-term outcome. Longer follow up for patient reported outcomes in future studies, is needed to determine whether the current less intense care pathway for older fallers with severe injuries is appropriate or represents therapeutic nihilism.”

---

## [Editor Report · Decision Letter 2]

19 Feb 2026

Comparative analysis of the characteristics, care pathways, and outcomes of English and Welsh major trauma patients injured by high versus low energy transfer mechanisms in 2019.

PONE-D-24-44855R2

Dear Dr. Lecky,

We’re pleased to inform you that your manuscript has been judged scientifically suitable for publication and will be formally accepted for publication once it meets all outstanding technical requirements.

Kind regards,

Kathleen Finlayson

Academic Editor

PLOS One

Additional Editor Comments (optional):

Thank you for addressing the reviewers' feedback
---

## [Editor Report · Acceptance letter]

PONE-D-24-44855R2

PLOS One

Dear Dr. Lecky,

I'm pleased to inform you that your manuscript has been deemed suitable for publication in PLOS One. Congratulations! Your manuscript is now being handed over to our production team.

Kind regards,

on behalf of

Dr. Kathleen Finlayson

Academic Editor

PLOS One